# Systematic Assessment of Protein C-Termini Mutated in Human Disorders

**DOI:** 10.3390/biom13020355

**Published:** 2023-02-12

**Authors:** Zachary T. FitzHugh, Martin R. Schiller

**Affiliations:** 1Nevada Institute of Personalized Medicine, University of Nevada Las Vegas, 4505 S. Maryland Pkwy, Las Vegas, NV 89154, USA; 2School of Life Sciences, University of Nevada, 4505 S. Maryland Parkway, Las Vegas, NV 89154, USA; 3Heligenics Inc., 833 Las Vegas Blvd. North, Suite B, Las Vegas, NV 89101, USA

**Keywords:** minimotifs, C-terminus, post-translational modification, acetylation, methylation, phosphorylation, bioinformatics

## Abstract

All proteins have a carboxyl terminus, and we previously summarized eight mutations in binding and trafficking sequence determinants in the C-terminus that, when disrupted, cause human diseases. These sequence elements for binding and trafficking sites, as well as post-translational modifications (PTMs), are called minimotifs or short linear motifs. We wanted to determine how frequently mutations in minimotifs in the C-terminus cause disease. We searched specifically for PTMs because mutation of a modified amino acid almost always changes the chemistry of the side chain and can be interpreted as loss-of-function. We analyzed data from ClinVar for disease variants, Minimotif Miner and the C-terminome for PTMs, and RefSeq for protein sequences, yielding 20 such potential disease-causing variants. After additional screening, they include six with a previously reported PTM disruption mechanism and nine with new hypotheses for mutated minimotifs in C-termini that may cause disease. These mutations were generally for different genes, with four different PTM types and several different diseases. Our study helps to identify new molecular mechanisms for nine separate variants that cause disease, and this type of analysis could be extended as databases grow and to binding and trafficking motifs. We conclude that mutated motifs in C-termini are an infrequent cause of disease.

## 1. Introduction

For rare diseases of large effect size, most of the mutations are within the coding regions [1]. Proteins are often composed of domains, with the average human protein having 2.06 domains [2]. These domains are collected in several databases, including PROSITE and the conserved domain database (CDD) [3,4,5]. The domains are often identified by motif signatures. They are also often identified by an algorithmic approach where sequence conservation is not consistent enough to define domains. Furthermore, some domain identities can only be determined after recognition of a common fold through 3D structure analysis [6].

Minimotifs are short functional sequence motifs that encode a function in a protein that is not a domain, such as post-translational modification (PTM) sites, binding sites, and trafficking elements [7,8]. The current datasets of Minimotif Miner (MnM) and Eukaryotic Linear Motif (ELM) have many consensus sequences and instances of these motifs [7,9]. An offshoot of the MnM database is the C-terminome database, which records over 3600 human proteins with a known minimotif in the last ten amino-acid residues [10]. This suggests that most or all human proteins may have functionalized C-termini.

What is not yet known is how often minimotifs are mutated in disease, though there are several reports that indicate that this is a disease mechanism [11,12,13]. In our first paper on the C-terminome, there were eight C-termini mutated in different diseases (Table 1) [14,15,16,17,18,19,20,21,22].

Collectively, these observations led us to question how many other C-termini may be mutated in diseases. In this paper, we used available databases to identify C-termini that are mutated in disease and the subset of those in which the mutation was in a known minimotif. We focused on PTMs because mutation of a key residue that is covalently modified changes the chemistry and can be reliably assigned as a loss-of-function (LOF). We designed a structured approach, taking advantage of current knowledge collected in several databases, to identify several known C-termini disease mechanisms and several new hypotheses where a PTM site is mutated in a disease but is not yet known to cause the disease.

## 2. Methods

Our bioinformatics analysis examined the presence of mutations in minimotifs from the C-terminome and MnM [7,8,10]. The C-terminome is a comprehensive database of over 3600 minimotifs in human and animal protein C-termini [10]. The MnM database likewise contains over one million motif sequences [7]. For mutation data, we used the NCBI ClinVar, an annotated repository of genetic variants that catalogues their pathogenicity and expressed phenotypes [23,24]. We implemented all data cleaning, analysis, and statistics with custom Python scripts. Most scripts relied on the pandas and numpy Python data science libraries.

Our first task was to procure the mutations from ClinVar and then join other datasets to the resulting table. First, we extracted all missense and nonsense SNPs from ClinVar (release date: May 2021) with the Python lxml library. We then obtained allele frequencies and GERP scores for the ClinVar mutations from gnomAD v2 Exomes and dbNSFP4.2 [25,26]. gnomAD frequencies were joined to ClinVar entries by chromosome, hg38 coordinates, reference nucleotide, and alternative nucleotide. We joined the dbNSFP information to the mutation data by these fields as well as the reference and alternative amino acid. We then constructed reference and alternative C-termini for each mutation using protein sequences and IDs from the NCBI RefSeq database [27]. Each ClinVar entry contains a set of RefSeq IDs designating the protein isoforms in which the mutation occurs. Using these RefSeq IDs as a set of keys, we joined the protein sequence data to the ClinVar entries. From there, we determined the alternative sequences and alternative C-termini caused by the mutations.

We subsequently prepared the minimotif data. First, we queried the MnM database for motifs on the C-termini. Second, we determined RefSeq IDs for the motif source proteins to facilitate a later join to the ClinVar mutations. Many table entries already had RefSeq IDs, but in some cases, motifs in the C-terminome and MnM data included only a UniProt ID for the source protein. Therefore, we acquired the corresponding NCBI RefSeq IDs for each entry in UniProtKB/SwissProt [28]. Then, we joined the RefSeq IDs to the motif entries by their listed UniProt IDs. When a RefSeq ID was linked to a non-human animal species, we matched the RefSeq ID to the human ortholog from the NCBI HomoloGene database where possible [29]. After determining the corresponding RefSeq IDs, we validated that each motif sequence was present in the C-terminus of the respective RefSeq protein. For that purpose, we used the RefSeq IDs to join the RefSeq sequence data to each motif. We then matched the motifs to the protein sequences with regular expressions. For motifs with PTM sites, we additionally verified that the amino acid of the PTM was present at the listed residue position. Lastly, we removed any duplicate motif entries between the C-terminome and MnM datasets.

In the final stage of our analysis, we compared the ClinVar mutation data and the motif data. The motif data were joined to the ClinVar data by RefSeq IDs. If a mutation occurred in the same protein as a motif, we determined if the mutation disrupted the motif via regular expression matching on the alternative C-terminus. We distinguished cases in which a mutation disrupted a minimotif without affecting the PTM site and cases where it directly substituted the site. We considered a nonsense mutation to disrupt a PTM site if the nonsense mutation occurred before the site residue. Our primary interest was in missense mutations that modified PTM sites, and we included an entry only if the ClinVar mutation was noted to be pathogenic or likely pathogenic. The mutations explored in our results are from this subset of the data.

We developed all figures using PowerPoint and the BioRender software tools. Figure 1 was developed in PowerPoint, whereas all remaining figures were created with BioRender. The graphical abstract was developed with a mixture of PowerPoint and BioRender.

## 3. Results

We designed a structured approach to screen for minimotifs in the C-termini that are mutated in human diseases (Figure 1). The first step in the flowchart was to parse missense and nonsense mutations from the ClinVar database [23,24]. We then extracted variant data from gnomAD and dbNSFP and matched it to the ClinVar data [25,26]. From there, we acquired proteome sequence data from the National Center of Biotechnology Information (NCBI) RefSeq database [27]. Next, we matched mutations to C-termini of each protein and created the alternative C-termini. We then parsed motif data from MnM and the C-terminome databases [7,8,10], using UniProt protein data in the process [28]. The motif data were matched to the disease mutation data to identify motif disruption mutations. We focused on PTMs because the amino acid that is covalently modified can confidently be assigned to be a LOF mutation when the chemistry is not possible. For example, substitution of a reference threonine with an alternative alanine necessarily prohibits phosphorylation. We also sorted out the subset of mutations classified as pathogenic or likely pathogenic. General statistics of our analysis are summarized in Table 2.

Our computational search rediscovered several known mutations in the C-termini that cause one or more human diseases. The filtered set had 20 variant instances across 16 PTM sites. In a secondary screen, we examined each instance in detail to determine if we identified a new hypothesis for a disease mechanism. The nine new instances are included in Table 3. In five other instances, a PTM site was substituted, but we discerned that site disruption is an unlikely disease mechanism. We also identified six instances in C-termini that had mutated PTMs already known in disease but not yet summarized in a mutated motif paper. These were in CSF1R, FUS, GluN2A, and rhodopsin, and the disease mechanisms are summarized below.

### 3.1. Rediscoveries of C-Terminal Minimotifs in Disease

#### 3.1.1. CSF1R

Our bioinformatics search identified several likely pathogenic mutations for hematologic neoplasms (HNs) on the C-terminus of colony stimulation factor 1 receptor (CSF1R), all at the same phosphorylated tyrosine residue [23,24,31]. HNs are tumors of white blood cells [32], and CSF1R is causative of these tumors [33,34,35,36,37,38]. The protein has a largely identical primary structure to the v-fms oncogene of feline sarcoma virus. In v-fms, the C-terminal 40 residues of CSF1R are substituted for an 11-residue sequence, and 9 other positions in the protein are replaced by missense variants [39]. The resemblance implies a negative regulatory function of the CSF1R C-terminus on tumorigenesis [40]. The mutations Y969F, Y969H, and Y969C at the final residue in the protein are likely pathogenic for HNs [23,24,41,42,43]. CSF1R autophosphorylates Y969 upon binding of CSF1. Phosphorylation subsequently allows c-CBL to bind to CSF1R and terminate CSF1R activity. The Y969F mutation abolishes c-CBL binding, inducing excessive, tumorigenic CSF1R activity [44]. The mutation also instigates cell transformation in mice cells [45].

#### 3.1.2. FUS

Another known C-terminal motif our analysis identified is a FUS mutation of a phosphorylation site that is causative for juvenile amyotrophic lateral sclerosis (JALS), a subtype of ALS. ALS is a neurodegenerative disease marked by fast deterioration of motor neurons that is typically lethal. Pathogenic FUS variants are especially prominent in JALS, and most occur in or near the nuclear localization signal in the C-terminus [46]. The pathogenic mutation Y526C is at the final residue within the nuclear localization signal [23,24,47]. Y526 is also a phosphorylation site that inhibits the FUS interaction with TNPO1, preventing nuclear translocation [31,48]. Therefore, the mechanism of JALS in Y526C is disruption of Y526 phosphorylation and its key role in FUS trafficking, leading to neurodegeneration.

#### 3.1.3. GluN2A

Our bioinformatics investigation identified a mutation in the C-terminus of GluN2A, a neurotransmitter receptor that is pathogenic for Landau–Kleffner syndrome (LKS), a notably rare and intense form of inherited epilepsy [49]. More than 30 mutations of GluN2A are pathogenic for epilepsy [50]. GluN2A is a key subunit in many NMDA receptors, which are involved in excitatory neurotransmission, synaptic plasticity, and memory formation [51]. The mutation S1459G in GluN2A is causative of LKS [23,24,52]. S1459 phosphorylation is an essential step in NMDA-receptor trafficking to the postsynaptic density, and serine binds to PSD-95 after translocation and dephosphorylation [31,53]. As such, disruption of the S1459 phosphorylation site in GluN2 inhibits NMDA trafficking and its interaction with PSD-95, causing LKS.

#### 3.1.4. Rhodopsin

The mutation S343N in a rhodopsin phosphorylation site in the C-terminus is likely pathogenic for retinitis pigmentosa [23,24], which induces progressive loss of vision over time. Deterioration of the rod cells of the retina causes the vision loss [54]. Expressed in rod cells, rhodopsin is a light receptor and transducer of the phototransduction cascade responsible for sight. After rhodopsin stimulation by light, retinal isomerizes, inducing conformational changes and phosphorylation of several threonine and serine residues in the C-terminus [55]. Phosphorylation of these residues, including S343, recruits arrestin and deactivates rhodopsin, terminating the signal [31,55,56,57,58,59,60,61]. In the absence of arrestin recruitment, excessive signaling causes neurodegeneration [58,62,63,64]. Therefore, S343N induces retinitis pigmentosa by disrupting a critical phosphorylation site and arrestin recruitment.

### 3.2. New Hypotheses for C-Terminal Minimotifs in Disease

In addition to minimotifs in the C-termini that were already mutated in disease, we also identified nine new instances where a PTM site was in the C-termini and mutated in a disease, but the connections were not previously recognized. Below, we summarize these new disease-causing hypotheses that will require further investigation to test their validity.

#### 3.2.1. Androgen Receptor

Androgen insensitivity syndrome (AIS) is a disorder caused by inhibited androgen signaling (Figure 2A). Patients are genetically male and carry the disorder as an X-linked trait. Symptoms can include infertility, undescended testes, excess testosterone, and a female appearance. Mutations in the androgen receptor are the major cause of the disorder [65]. Androgens bind to residues in the C-terminal ligand-binding domain of the androgen receptor. The receptor then translocates to the nucleus, binds to the hormone response element (HRE) element in the promoter, and initiates transcription of genes integral to male sexual development. Androgen receptor mutations can disrupt this process, leading to AIS [66].

Cleaving the last 12 C-terminal residues of the androgen receptor eliminates binding of agonists and antagonists, indicating that the C-terminus is critical for function [68]. Likewise, missense mutations in the region cause AIS [69]. Site-directed mutagenesis of 10 of the last 13 residues identified that all but three of the 10 mutated residues contribute to ligand binding [70].

Y915 is one of the androgen receptor residues that affect ligand binding [70], and Y915S is causative of AIS [23,24]. Our bioinformatics analysis identified the Y915S mutation and phosphorylation of Y915 by Src kinases [31,67]. In fact, when the androgen receptor is tyrosine-phosphorylated, it is activated in an androgen-independent manner. It is not certain that Y915 is the key site because there are 10 other phosphorylated tyrosine sites throughout the protein [67]. Additionally, it is not yet known if androgen-independent activation is abolished by mutation of Y915. Given the pathogenic nature of Y915S, we propose the hypothesis that mutation of Y915 completely blocks Src kinase phosphorylation of this site and induces AIS by inhibiting androgen-independent activation (Figure 2B). Alternatively, Y915S may just block androgen binding. Further experimentation is needed to test these hypotheses.

#### 3.2.2. APOC-III

Apolipoprotein C-III (APOC-III) is a key constituent of HDL, LDL, and VLDL, transporting lipids through the bloodstream. It additionally inhibits the activity of lipoprotein lipase and the uptake of triglycerides by hepatocytes [71]. Elevated bloodstream APOC-III is associated with cardiovascular diseases, hypertriglyceridemia, and diabetes [72,73].

The T74A mutation in the C-terminus of APOC-III abolishes its principal glycosylation site (Figure 3) [74]. The mutation was first observed in a normolipidemic, heterozygous Japanese family and is reported in ClinVar [23,24,75]. Most APOC-III molecules are glycosylated, where an O-linkage consisting of a single galactose molecule and a single N-acetylgalactosamine molecule is established (Figure 3) [76]. Glycosylation and sialylation of the O-linkage distinguish four glycoforms: APOC-III_0A_ is non-glycosylated, APOC-III_0B_ is non-sialylated, APOC-III_1_ is monosialylated, and APOC-III_2_ is disialylated [77].

Several studies have observed a correlation between a greater degree of APOC-III sialylation and reduced risk of atherosclerosis, as well as lower plasma triglyceride levels [77,79,80,81,82,83,84,85]. Others have reported the opposite effect: a link between increased APOC-III sialylation, hypertriglyceridemia, and diabetic hemodialysis [86,87,88,89]. Nevertheless, the function of APOC-III glycosylation is not yet fully understood, and there is conflicting evidence regarding its effect on disease. For example, an in vitro experiment expressing APOC-III with T74A indicated that the mutation does not inhibit APOC-III secretion and association with VLDL [78].

Our computational study revealed phosphorylation of T74 in APOC-III [31]. We expect that this PTM competes with T74 O-glycosylation (Figure 3). Therefore, we hypothesize that T74 phosphorylation blocks APOC-III glycosylation and sialylation and may affect lipid metabolism and lipidemic diseases. Studies of the site’s phosphorylation and how it affects glycosylation and lipid particle physiology may clarify the pathology of APOC-III-related diseases.

#### 3.2.3. CRYM

Nonsyndromic sensorineural deafness (NSD) is deafness resulting from dysfunction of the inner ear that is not symptomatic of an underlying syndrome. There are several genes associated with NSD, with one being µ-Crystallin. A screen of 192 Japanese families with NSD discovered causative mutations in µ-Crystallin [90]. Moreover, a Chinese family with NSD inherited an N-terminal P51L mutation, further supporting a role for µ-Crystallin in NSD pathology [91]. Thyroid hormone binds µ-Crystallin, stimulating nuclear translocation and activation of a thyroid-hormone signaling pathway. When µ-Crystallin binds NADP^+^ and then to T_3_, it is translocated to the nucleus (Figure 4A). Alternatively, when µ-Crystallin binds the reduced form of NADP^+^, NADPH, translocation of the complex with T_3_ is inhibited (Figure 4B) [92].

Like the trafficking role of many C-termini [14], mutations in the C-terminus of µ-Crystallin interfere with its subcellular localization, which is critical for its thyroid-hormone signaling function. Abe et al. reported two variants in NSD patients in the C-terminus of µ-Crystallin. A stop loss mutation X315Y extends the C-terminus by five residues and is mislocalized to vacuoles in transfected COS-7 cells. Similarly, a K314T missense mutation is mislocalized perinuclearly. These mutations may also affect potassium ion recycling stimulated by thyroid-hormone activation of the Na,K-ATPase ion transporter [90].

The causative K314T mutation in the last residue of µ-Crystallin (YDSWSSGK>) is pathogenic for NSD [23,24,90]. The homologous residue in mouse µ-Crystallin is acetylated [31], suggesting our hypothesis that loss of K314 acetylation is pathogenic (Figure 4C). K314 acetylation is not yet linked to disease pathology. Disruption of K314 acetylation and the K314T mutation may contribute to NSD etiology by three possible mechanisms. First, the subcellular mislocalization established by Abe et al. may impede thyroid-hormone trafficking. Second, potassium-ion recycling may be disrupted, negatively impacting hearing function. Third, µ-Crystallin with the K314T mutation cannot bind to T_3_ at all [93], which eliminates nuclear translocation of thyroid hormone by µ-Crystallin. These three mechanisms could potentially work in concert with each other. Additional work will be needed to clarify the role of K314 acetylation in these processes and the etiology of NSD.

#### 3.2.4. GMPPB

Muscular dystrophy dystroglycanopathy (MDDG) is a major class of muscular disease. In MDDG, insufficient activity of α-dystroglycan weakens the stability of muscle and brain cell membranes [94,95]. The GDP-mannose pyrophosphorylase B (GMPPB) gene is causative of the disorder [94,96]. In zebrafish models, sufficient GMPPB enzymatic activity is necessary for normal neuronal and muscular development [97]. GMPPB is part of a complex that catalyzes the mannose-1-phosphate and GTP substrate conversion to the GDP-mannose product (Figure 5A) [98]. GDP-mannose reacts with Ser or Thr hydroxyl groups, O-mannosylating α-dystroglycan. α-dystroglycan interaction with extracellular proteins is dependent on its mannosylation. Thus, dysfunctional GMPPB activity induces MDDG through this pathway [94,99,100].

GMPPB mutations also cause a related disease called congenital myasthenic syndrome (CMS). CMS constitutes a distinct spectrum of neuromuscular disease, characterized by dysfunction of the neuromuscular joint [101]. GDP-mannose is a key precursor of N-glycans [94]. Deficient GDP-mannose production, therefore, likely impairs the glycosylation necessary for neuromuscular communication, instigating CMS. These two diseases have similar mechanisms involving GMPPB mutations, thus the proposed mechanism may also be related to CMS.

In total, there are 21 disease-associated variants of GMPPB in its C-terminal domain that are essential for interaction with the C-termini of other subunits in the catalytic GDP-mannose pyrophosphorylase complex. The complex consists of several GMPPA and GMPPB subunit homodimers and heterodimers. Therefore, C-terminal GMPPB mutations likely impair complex formation, producing MDDG pathology [98].

The mutation R357H in GMPPB’s C-terminus was identified in five Chinese patients afflicted with MDDG and CMS [23,24,95]. These results suggest that the R357H mutation is likely pathogenic for MDDG and CMS. R357H, like other MDDG missense mutations, is in the C-terminus of GMPPB that is critical for assembly of the catalytic complex.

Our bioinformatics investigation identified a separate study where R357 is methylated [31]. There are three possibilities: that methylation has no effect, promotes, or inhibits formation of the catalytic complex. Given that the mutation is likely pathogenic and that disrupted GMPPB activity is pathogenic, we hypothesize that R357 methylation is required for complex formation and that the R357H mutation inhibits methylation and contributes to muscular disorder etiology (Figure 5B). Further experimentation should be conducted to solidify the connection between R357 methylation and muscular dysfunction.

#### 3.2.5. Hemoglobin

Secondary erythrocytosis is a disease characterized by abnormally high red blood cell (RBC) count [102]. Hemoglobin A, the principal form of hemoglobin in adult RBCs, is a heterotetramer composed of two hemoglobin alpha (HBA) and two hemoglobin beta (HBB) subunits [103]. Mutations in HBA and HBB are causative of erythrocytosis [104]. Over 100 HBB and HBA mutations increase the oxygen affinity of hemoglobin, often causing erythrocytosis. Many such mutations are concentrated in the C-termini [105,106].

In its unliganded T (tense) state, salt bridges on the C-termini stabilize hemoglobin A’s heterotetrameric structure, which is made up of two HBB-HBA heterodimers (Figure 6A) [107,108,109]. As oxygen molecules bind to hemoglobin, critical interdimeric α_1_β_2_ contacts rupture when a penultimate tyrosine residue on each chain is displaced from pockets between the F and H helices. The new conformer is called the R (relaxed) state. Additionally, the final residue in each HBA subunit forms salt bridges with the HBA subunit of the opposing dimer [109]. The penultimate tyrosine residues in HBB and HBA also participate in hydrogen bonds in the T state that are lost in the R state [109,110]. Therefore, hemoglobin mutations in the C-terminus destabilize the T state and favor the R state, raising its affinity for oxygen [104,111,112,113,114,115,116,117]. The consequent increase in oxygen affinity inhibits the release of oxygen to the tissues, provoking tissue hypoxia. The kidneys compensate by elevating erythropoietin secretion, stimulating increased RBC count [102,104].

The mutations Y145H (Hemoglobin Bethesda), Y145C (Hemoglobin Rainier), and Y145N (Hemoglobin Osler) in HBB induce erythrocytosis [23,24,118,119,120,121,122,123,124]. Residue Y145 corresponds to the penultimate residue of HBB and is also a phosphorylation site [31]. The coincidence of these mutations with a PTM site suggests our hypothesis that elimination of Y145 phosphorylation disrupts regulation of the R-to-T-state transition and causes erythrocytosis (Figure 6B). In addition to the role played by the tyrosine residues in interdimeric bonds, we propose that phosphorylation of Y145 contributes to the stability of the T-state heterotetramer. Therefore, inhibiting phosphorylation favors the R state and elevates oxygen affinity. Further experimental work should be pursued to test this hypothesis.

The mutations K144N (Hb Andrew-Minneapolis) in HBB and K139E (Hb Hanamaki) in HBA1 also produce erythrocytosis [23,24,125,126,127,128]. The residues K144 in HBB and K139 in HBA are acetylated [31]. However, studies of hemoglobin acetylation by aspirin, including K144, demonstrate either no effect on oxygen affinity or increased oxygen affinity [129,130,131]. Therefore, we do not believe that disrupted acetylation is a major factor in erythrocytosis etiology. Instead, these mutations likely cause disease by some other mechanism, such as destabilizing T-state salt bridges. We additionally did not consider the mutation Y140H (Hemoglobin Ethiopia/Rouen) in HBA, which substitutes a phosphorylated residue, since its probands displayed higher oxygen affinity but not compensatory erythrocytosis [23,24,30,31,132]. Given the lack of evidence of PTM disruption as a disease mechanism, we have included these three mutations in the “unlikely” category in Table 3.

#### 3.2.6. SDHA

Defective mitochondrial complex II activity produces dysfunctional metabolism, causing mitochondrial complex II deficiency (MCIID). MCIID is a rare disease, accounting for 2% of patients with mitochondrial disease, and can cause neuropathies and other diseases [133,134]. Succinate dehydrogenase A (SDHA) is a subunit of mitochondrial complex II (SDH), a key enzyme in the tricarboxylic acid (TCA) cycle and oxidative phosphorylation (Figure 7A) [133]. In addition to its role in cell metabolism, the SDH complex operates as a tumor suppressor [135].

Seventeen pathogenic variants of SDHA cause MCIID, and SDHA mutations can also induce familial paragangliomas and pheochromocytomas [135]. R662C is likely pathogenic for MCIID and was additionally present in a patient with a paraganglioma [23,24,139]. This position has multiple roles, affecting binding to the flavin adenine dinucleotide (FAD) cofactor, forming a salt bridge in a complex intermediate with the dedicated assembly factor subunit SDHAF2, and potentially contributing to the stability of the SDH complex [136,139,140].

Our bioinformatics analysis discovered that R662 is methylated [31]. We hypothesize that inhibited R662 methylation in the R662C mutant disrupts SDHA flavinylation (Figure 7B), complex maturation (Figure 7C), and/or SDH complex stability (Figure 7D), contributing to MCIID pathogenesis and paragangliomas. Further experimentation should be pursued to assess the impact of R662 methylation on these processes.

#### 3.2.7. SMAD3

Familial thoracic aortic aneurysmal disease (FTAAD) is an inherited propensity for the development of aortic aneurysms. A thoracic aortic aneurysm occurs when the diameter of the thoracic aorta enlarges to 150% of its normal size. In extreme cases, an aneurysm precipitates dissection of the aorta [141]. Related conditions include Marfan syndrome and Loeys–Dietz syndrome [142].

Mutations of SMAD3 cause FTAAD in 2% of cases and are also causative of Marfan syndrome and Loeys–Dietz syndrome [143]. This is supported by studies of SMAD3 knockout mice, which experience greatly accelerated development of aortic aneurysms and death [144]. A 2018 study compiled 67 SMAD3 pathogenic and likely pathogenic variants associated with Loeys–Dietz syndrome [145], and a 2019 study of patients with SMAD3 mutations found that more than half experienced dissections [146].

SMAD3 is a key transducer of the TGF-β signaling pathway, and its C-terminus is critical for signal transduction (Figure 8A). Disruption of the pathway instigates aortic aneurysms. Upon activation, TGF-β binds to TβRII and TβRI receptors. TβRII then phosphorylates TβRI. Subsequently, TβRI phosphorylates two of the C-terminal serine residues in the SSXS> motif of SMAD2 or SMAD3. The activated SMAD2 or SMAD3 forms a complex with co-SMADs. The resulting complex translocates to the nucleus, where it regulates gene transcription [147].

Our analysis identified the likely pathogenic mutation S423N in the SSXS> motif of the C-terminus, along with S423 phosphorylation [31]. It was reported in ClinVar in two individuals with FTAAD, one of which also suffered Marfan syndrome [23,24]. Moreover, the aforementioned 2018 study also documented this mutation in Loeys–Dietz syndrome [145]. However, neither study recognized that this mutation disrupted this key signaling motif. Therefore, we hypothesize that this variant blocks SMAD3 phosphorylation and disrupts the TGF-β signaling pathway, resulting in disease of the aortic wall (Figure 8B).

### 3.3. Unlikely Hypotheses for C-Terminal Minimotifs in Disease

#### 3.3.1. APOE

APOE is a constituent molecule of lipoproteins such as chylomicrons, HDL, LDL, and VLDL. It is widely present in the central nervous system, where it is secreted by astrocytes [148]. Possession of the APOE4 isoform is the greatest risk factor for Alzheimer’s disease, and APOE dysfunction is related to coronary heart disease [149]. Our study identified an O-glycosylation and phosphorylation site at S296 that is abolished by an S296R mutation [23,24,31,148,150,151,152]. The function of O-glycosylation in APOE is disputed, with some studies linking greater glycosylation to superior health outcomes and others implying the opposite [79,153,154,155,156]. The protein kinase CK2 preferentially phosphorylates S296 in vitro. However, phosphorylation likely does not greatly impact APOE function in vivo since most APOE is lipidated in neurons and since there are no known kinases in the secretory pathway [152]. Similarly, disruption of APOE O-glycosylation does not impede secretion [157,158], and the mutation was found in a normolipidemic proband [150].

#### 3.3.2. INS

Monogenic mutations are causative in about 1–2% of diabetes cases, and mutations in the *INS* gene are common [159]. The insulin mutation Y108C is pathogenic for congenital neonatal diabetes and juxtaposed to a disulfide bridge between the A and B chains [23,24]. Our bioinformatic analysis identified phosphorylation of Y108, which is also on the C-terminus (YCN>) [31]. While disruption of Y108 phosphorylation could contribute to insulin activity and cause congenital neonatal diabetes, insulin is very well-studied, there are no known tyrosine kinases in the lumen of the secretory pathway, and insulin is not known to be tyrosine phosphorylated in vivo. Therefore, Y108 phosphorylation is not a likely mechanism in congenital neonatal diabetes etiology.

## 4. Discussion

Our analysis of PTMs in protein C-termini that are mutated in disease led to several conclusions:Twenty missense mutations in PTM sites are reported to cause human disease.We developed novel hypotheses for nine mutations in seven genes and PTM sites where disruption of a PTM causes a human disease.The analysis of available databases for disease shows that the disruption of C-terminal PTMs, while present, is far less frequent than missense mutations throughout coding regions.

The mutation rate of our dataset estimates the prevalence of pathogenic mutations in PTM sites in protein C-termini. There are roughly 19,800 protein-coding genes in the human genome [160]. Considering the 20 mutations identified from our study of the C-termini in the human proteome (10 amino acids), the mutation rate is $\frac{20}{10*19800}=0.0001$, or roughly 0.01%. Eliminating the 5 mutations in the “unlikely” category, the rate is instead about 0.008%. We can contrast this metric with the overall pathogenic missense mutation rate across the entire human proteome. There are currently 50,001 pathogenic and likely pathogenic missense variants in ClinVar [23,24], and there are at least 10.8 million amino acid residue positions in the human proteome [161]. This gives an estimated mutation rate of roughly 0.5%, though the rate is based on current knowledge and is possibly much higher. The overall pathogenic missense mutation rate is thus approximately 63 times higher than the pathogenic mutation rate of C-terminal PTM sites. These results suggest that PTM sites in the C-terminal region are highly conserved relative to the rest of the proteome and that these mutations are rare. However, the total rate of pathogenic mutations in minimotifs may be underestimated since we did not assess binding or trafficking motifs because we cannot conclusively assign an LOF mutation for most minimotifs.

All 20 of the reported mutations were classified as rare, either with a low frequency as estimated in gnomAD or not appearing at all. If a mutation was absent in gnomAD, we assigned it a frequency of <7.95 × 10^6^ in Table 3. Genome evolutionary rate profiling (GERP) scores were used to assess the conservation of each mutation given that a highly conserved position is more likely to be disease-causing [162]. All but 4 mutations had a GERP above score above 2, with these 4 split among the unlikely category (n = 2), new hypothesis category (n = 1), and the rediscovery category (n = 1). The results of low frequencies and high GERP scores for most mutations that we identified are consistent with their pathogenic character. 

We also assessed potential pathogenicity with scores from commonly used algorithms (SIFT, SIFT4G, PolyPhen-2 HumDiv, PolyPhen-2 HumVar, MPC, and CADD); see Appendix A [163,164,165,166,167,168,169]. Scores were acquired from dbNSFP [25]. Most scores were consistent with the mutations being damaging. Only three mutations were predicted to be ”tolerated” by SIFT. Similarly, SIFT4G predicted only two to be “tolerated.” Thus, 85% and 90% of mutations were predicted to be damaging by SIFT and SIFT4G, respectively. Three mutations had no PolyPhen-2 predictions in dbNSFP. Of the mutations with PolyPhen-2 scores, approximately 76.5% and 58.8% were predicted to be damaging by PolyPhen-2 HumDiv and PolyPhen-2 HumVar, respectively. CADD PHRED-scaled scores were above the threshold of 20 for deleteriousness for 85% of mutations. On the other hand, most MPC scores were somewhat low on the 0 to 5 scale, with all but 2 mutations having a score below 2. The results from SIFT, SIFT4G, PolyPhen-2 HumDiv, and CADD support the pathogenic nature of the mutations in our study, though the MPC scores and PolyPhen-2 HumVar scores are less favorable.

Our analysis of C-terminal minimotifs in disease has several limitations. We only analyzed PTMs, and not binding or trafficking motifs, because of the simpler interpretation with reliable assignment of LOF alleles. Moreover, we only analyzed the PTM site itself rather than other motif positions such as the S or T in the Nx(S/T) N-glycosylation site because the requirement for these positions is less certain. Furthermore, we had previously identified 8 C-terminal binding and trafficking sites that cause human disease; thus, a future study could investigate these other classes. The challenge is that the motif datasets would require annotation for which residues are required, and this is not yet available in the MnM and ELM databases. An additional limitation is that not all disease-causing mutations are in ClinVar.

We would like to conclude that mutation of C-terminal motifs is not a significant disease mechanism. Our data suggest that this is the case, with only dozens of causative mutations. However, the variant and motif databases we used are not yet comprehensive. Nevertheless, our investigation suggests that C-terminal motifs that are mutated and cause disease do occur but have low prevalence.

## Figures and Tables

**Figure 1 biomolecules-13-00355-f001:**
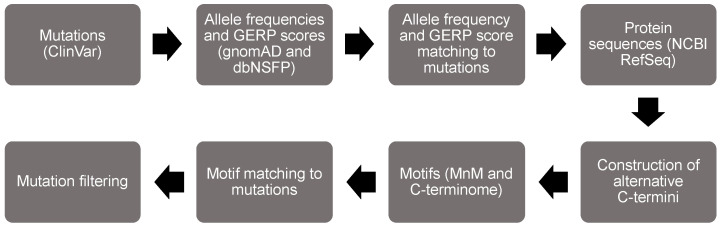
Flow chart of the bioinformatic analysis to identify minimotifs in the C-termini that are mutated in disease.

**Figure 2 biomolecules-13-00355-f002:**
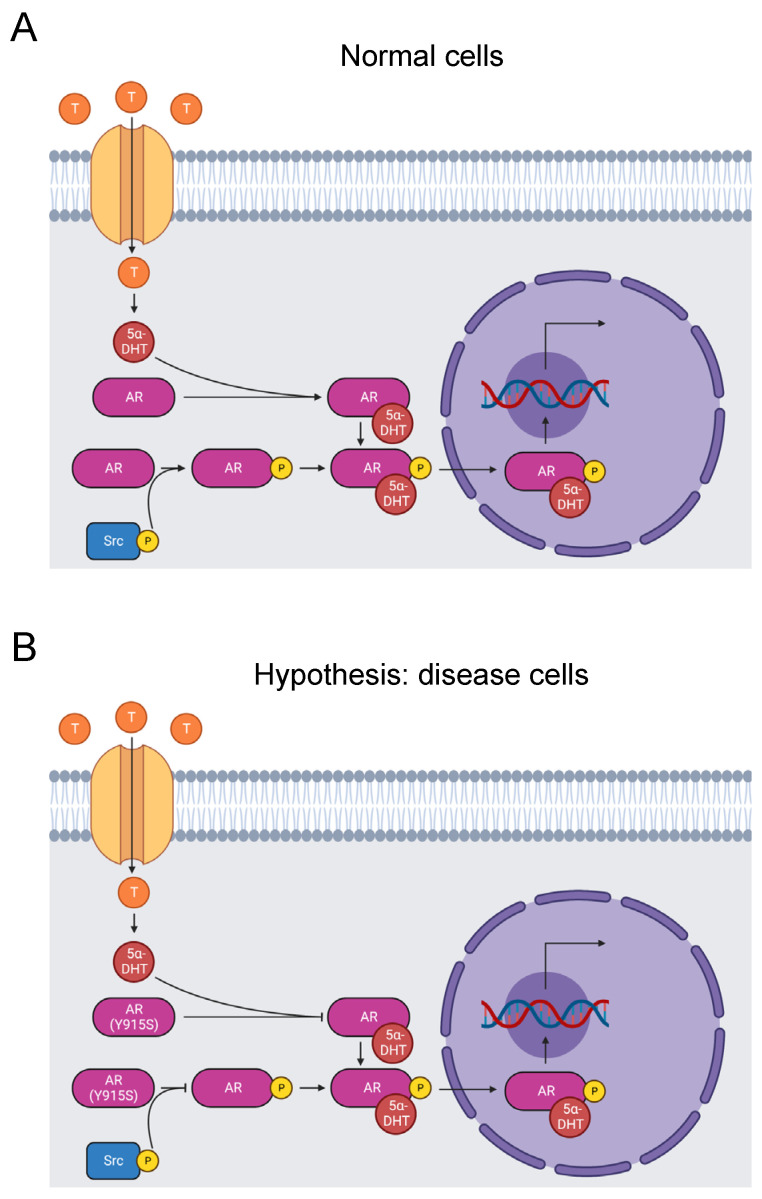
Diagram of the AR signaling pathway. (**A**) Testosterone enters prostate cells, where it converts to 5α-dihydrotestosterone. It binds to and activates the AR, which is phosphorylated on its ligand-binding domain by a Src kinase. The AR translocates to the nucleus and stimulates RNA transcription that promotes male sexual development [66]. Tyrosine phosphorylation by Src kinases can also activate the receptor independent of androgen binding [67]. (**B**) The mutation Y915S may cause AIS by abolishing tyrosine phosphorylation of the residue and disrupting androgen-independent AR activation. Alternatively, Y915S causes AIS by inhibiting androgen binding to the AR [23,24].

**Figure 3 biomolecules-13-00355-f003:**
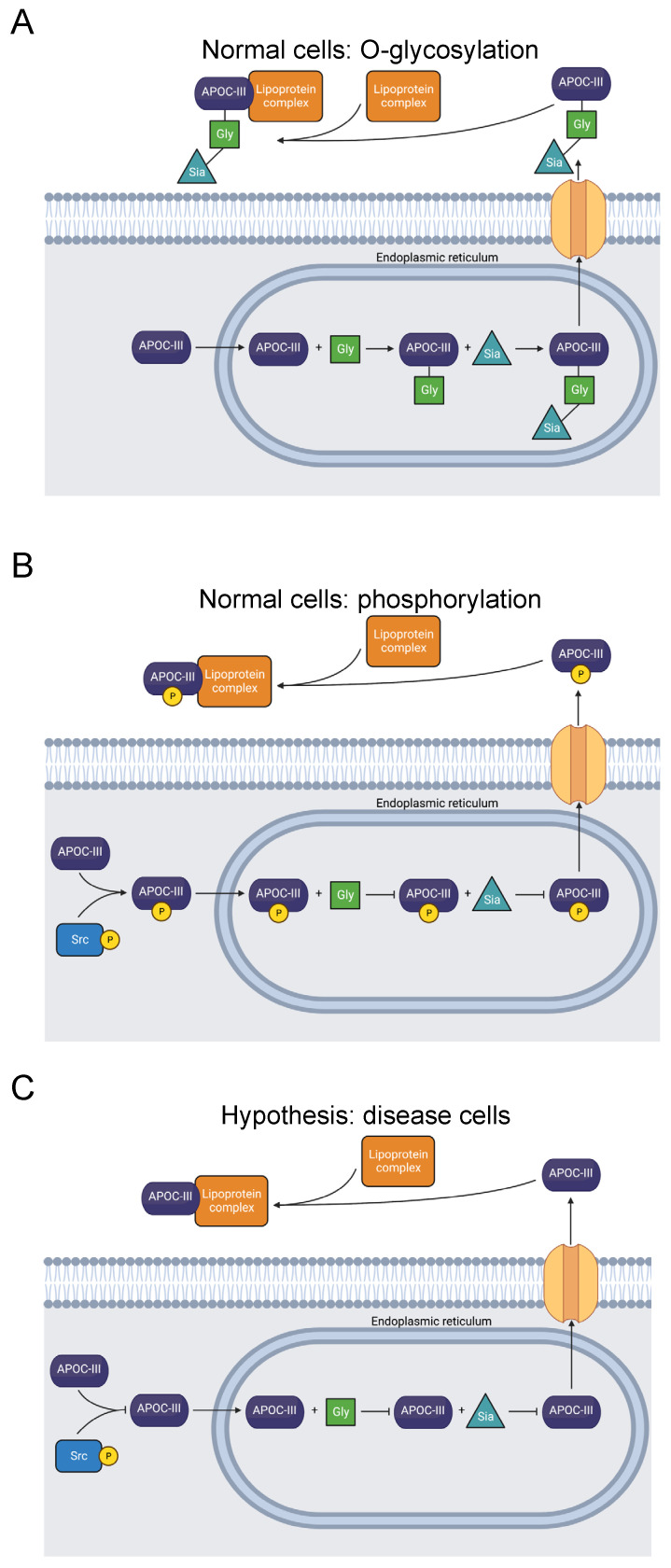
Diagram showing O-glycosylation, sialylation, phosphorylation, and extracellular transport of APOC-III. (**A**) APOC-III is O-glycosylated and sialylated in the endoplasmic reticulum [76,77]. After extracellular transport, it becomes a constituent of lipoproteins, regulating lipid transport and metabolism [71]. (**B**) Alternatively, T74 is phosphorylated, precluding O-glycosylation [31]. (**C**) The mutation T74A abolishes the O-glycosylation and phosphorylation site [23,24,74,75]. Inhibited O-glycosylation and phosphorylation of APOC-III may be related to dysfunctional lipid metabolism, though secretion is not prevented [78].

**Figure 4 biomolecules-13-00355-f004:**
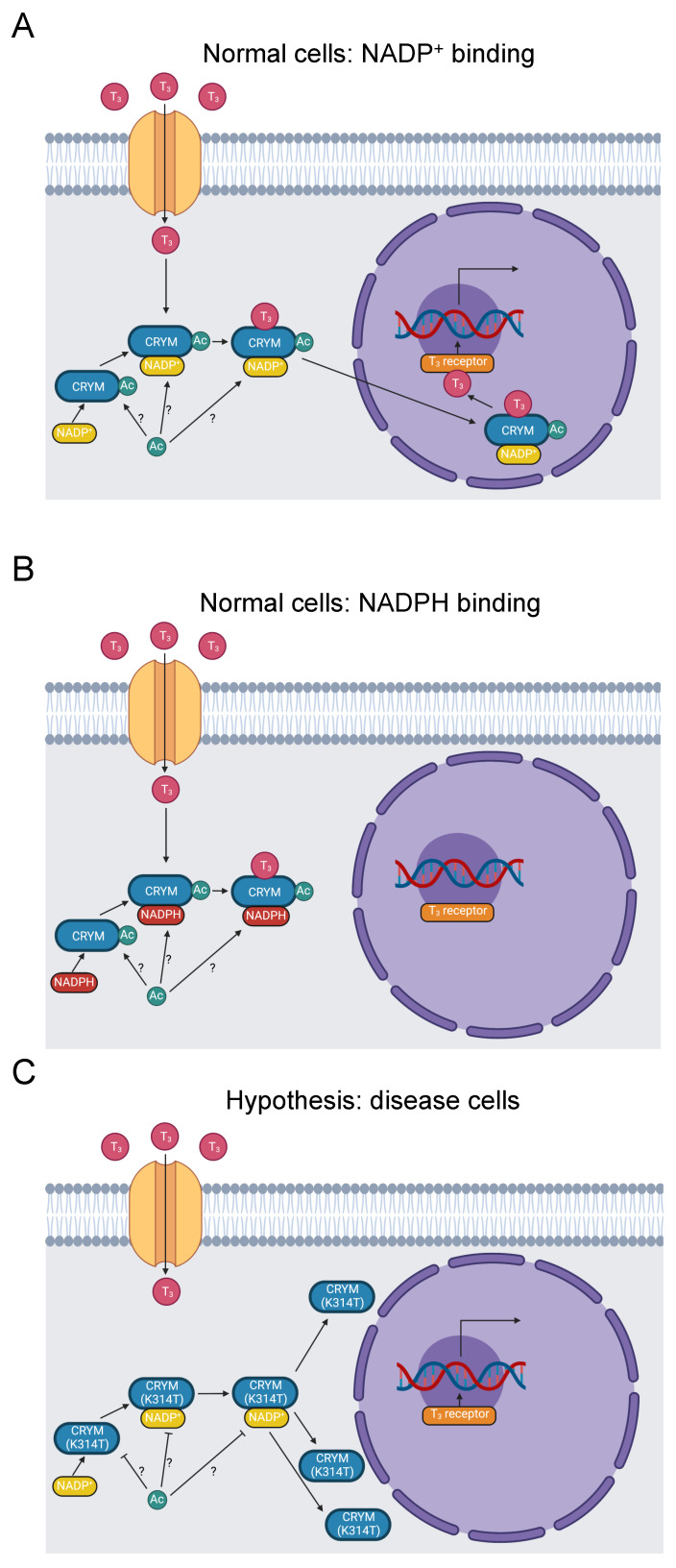
Diagram of thyroid-hormone trafficking by μ-Crystallin and subsequent transcription promoted by a thyroid-hormone receptor. (**A**) Thyroid hormone enters the cytoplasm and binds to μ-Crystallin in the presence of NADP^+^. The complex translocates to the nucleus, where thyroid hormone binds to a receptor and stimulates RNA transcription [92]. K314 is acetylated [31]. Whether this occurs when μ-Crystallin is free, when it binds to NADP^+^, or when it binds to thyroid hormone is indeterminate, as indicated by the “?” symbol. (**B**) When NADPH binds to μ-Crystallin instead of NADP^+^, the complex fails to translocate to the nucleus [92]. (**C**) The K314T mutation abolishes thyroid-hormone binding to μ-Crystallin and results in a perinuclear distribution of μ-Crystallin [90,93]. It, therefore, prevents thyroid-hormone signaling downstream of μ-Crystallin binding. It further inhibits acetylation of K314.

**Figure 5 biomolecules-13-00355-f005:**
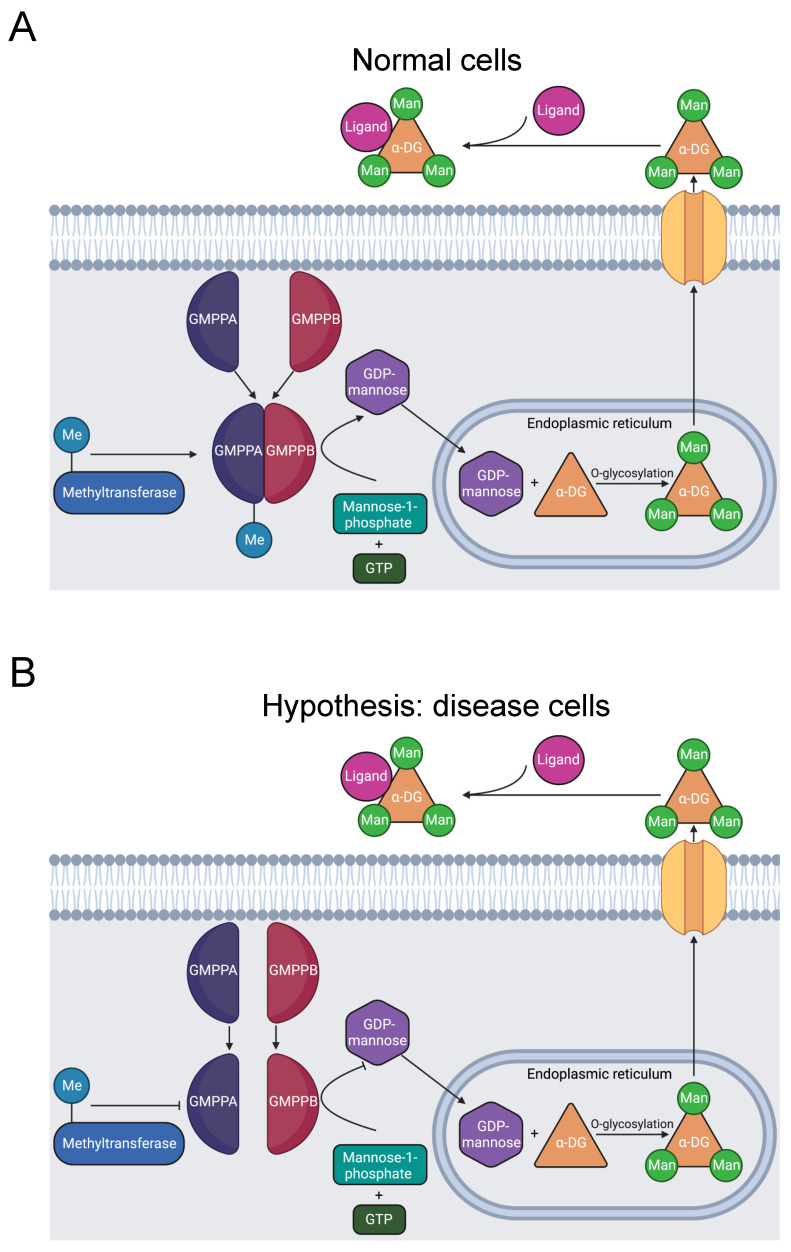
Diagram of GMPPB catalysis of α-DG O-mannosylation. (**A**) GMPPA and GMPPB subunits form a stable complex, including methylation of R357 [31]. The complex catalyzes the reaction of GTP and mannose-1-phosphate that produces GDP-mannose. In the endoplasmic reticulum, GDP-mannose O-mannosylates α-DG. Once transported to the extracellular matrix, α-DG binds to ligands and stimulates downstream signaling pathways [94]. (**B**) R357H inhibits R357 methylation and causes MDDG [23,24,95], likely by preventing stable complex formation [98]. The resulting instability reduces the quantity of GDP-mannose produced and decreases O-mannosylation of α-DG, causing MDDG [94,99,100].

**Figure 6 biomolecules-13-00355-f006:**
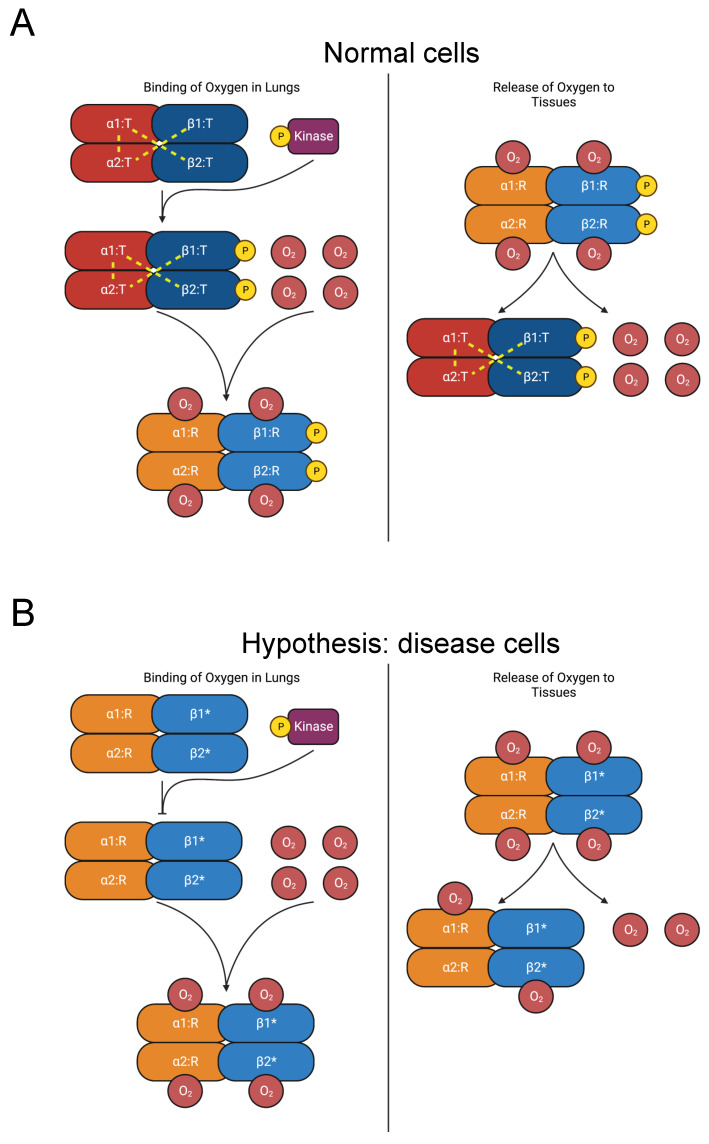
Diagram of oxygen binding to hemoglobin and subsequent release to tissues. (**A**) In the lungs, oxygen binds to hemoglobin, which is composed of two αβ heterodimers [107,108,109]. Y145 on the β chains is phosphorylated [31]. Interdimeric bonds that stabilize the T state are broken as oxygen binds, and the molecule transitions to the R state, increasing oxygen affinity [109,110]. In the tissues, oxygen is released, and hemoglobin reverts to the T state. (**B**) Mutations of Y145 (such as Y145H, Y145C, and Y145N, indicated by “β1*” and “β2*”) disrupt interdimeric bonds and phosphorylation, favoring the R state over the T state and increasing oxygen affinity [104,111,112,113,114,115,116,117]. The consequent increase in oxygen affinity diminishes the release of oxygen to the tissues and provokes compensatory erythrocytosis [102,104].

**Figure 7 biomolecules-13-00355-f007:**
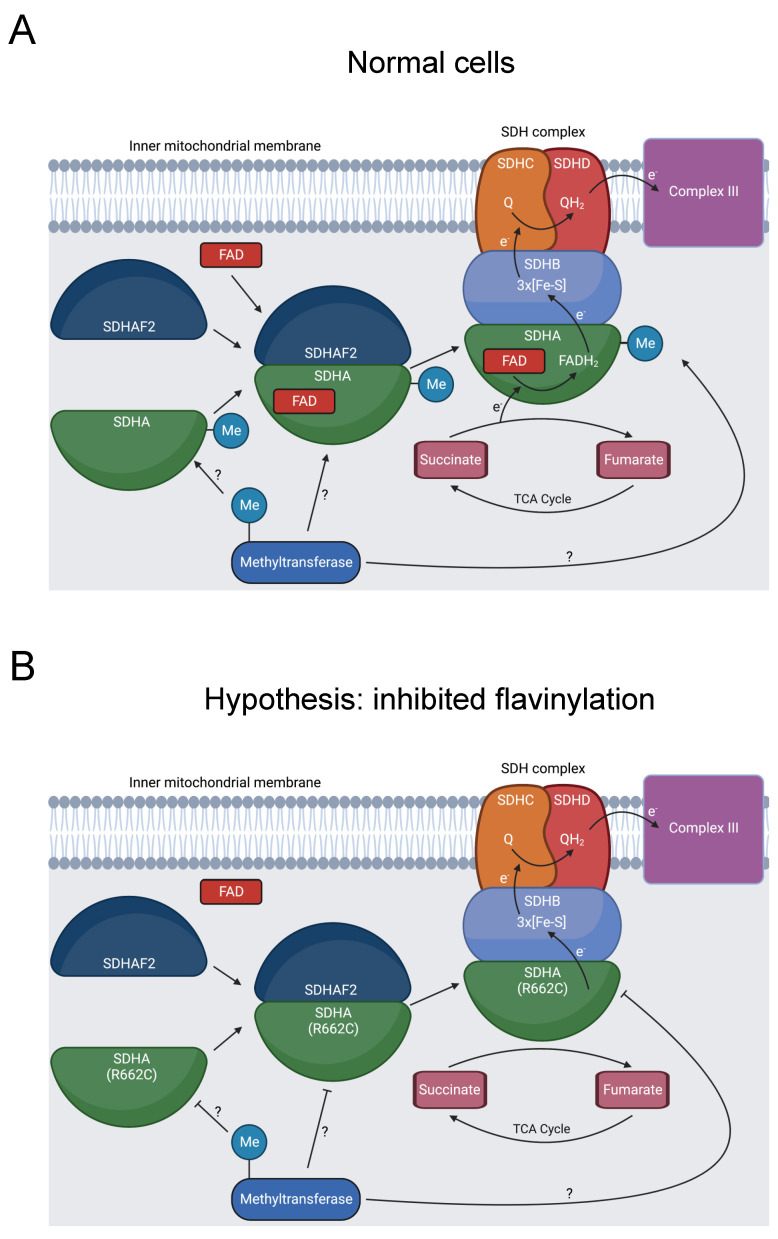
Diagram showing formation of the SDHA-SDHAF2 complex, SDHA methylation, and SDH’s role in oxidative phosphorylation and the TCA cycle. (**A**) SDHA and SDHAF2 form an intermediate complex, which promotes binding of the FAD cofactor to SDHA [136,137]. The full SDH complex then assembles. The heterotetramer operates as complex II in the electron transport chain and oxidizes succinate to fumarate in the TCA cycle [138]. At some point, R662 of SDHA is methylated [31]. This could occur prior to the SDHA-SDHAF2 complex formation, during SDHA-SDHAF2 complex formation, during SDH complex assembly, or after SDH complex assembly, as indicated by the “?” symbol. (**B**) R662C disrupts R662 methylation and precipitates MCIID [23,24,139]. A possible mechanism is disruption of SDHA flavinylation. (**C**) Another possible mechanism is inhibited complex maturation and formation of the SDHA-SDHAF2 complex. (**D**) A final proposed mechanism is increased instability of the full SDH complex.

**Figure 8 biomolecules-13-00355-f008:**
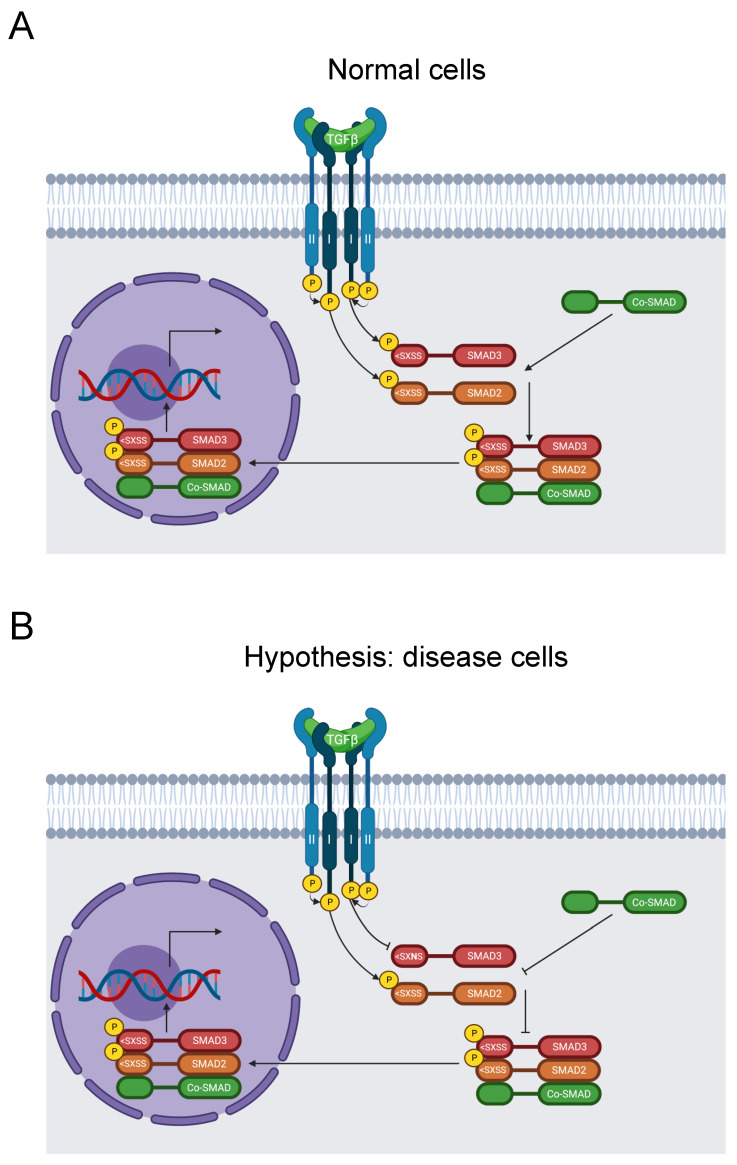
Diagram of the SMAD signaling pathway. (**A**) TGF-β binds to TβRII and subsequently recruits TβRI. TβRII subsequently activates TβRI by phosphorylation, which transduces the signal by phosphorylating SMAD2 and SMAD3 on their SXSS > motifs. Activated SMAD2 and SMAD3 then form a complex with a co-SMAD before translocating to the nucleus and stimulating gene expression [147]. (**B**) The S423N mutation disrupts the key SSXS > motif for SMAD3 phosphorylation and activation, thereby inhibiting further signal transduction and regulation of RNA transcription by the SMAD complex [147]. This causes FTAAD [23,24].

**Table 1 biomolecules-13-00355-t001:** Eight known C-terminal motifs in disease.

Protein	Minimotif	General Function	Disease	Reference
K_v_11.1	RGRX_152_> ^1^	Trafficking	Long QT syndrome type 2	Kupershmidt et al. (2002), [15]
Rhodopsin	VAPA>	Trafficking	Vision loss	Deretic et al. (2005), [16]
MNK	LLX_12_> ^1^	Trafficking	Menkes disease	Petris et al. (1998), [17]
NKCC2	LLX_13_> ^1^	Trafficking	Bartter syndrome	Zaarour et al. (2012), [18]
BSEP	YYKLVX_7_> ^1^	Trafficking	Primary familial intrahepatic cholestasis type 2	Lam et al. (2012), [19]
SANS	TEL>	Binding	Usher syndrome type 1	Reiners et al. (2006), [20]
Claudin-16	TRV>	Binding	Nephrocalcinosis	Müller et al. (2003), [21]
NaV1.5	SIV>	Binding	Cardiac disease	Shy et al. (2013), [22]

^1^ The symbol “X_y_>” indicates that y additional amino acids occur before the C-terminus.

**Table 2 biomolecules-13-00355-t002:** ClinVar mutation statistics.

Category	Count
Total ClinVar variants	1,260,173
ClinVar nonsense and C-terminal missense variants	37,046
ClinVar C-terminal missense variants	5550
Pathogenic variants with PTM disrupted	20
Uncertain variants with PTM disrupted	119
Total with PTM disrupted	213
Pathogenic with motif but not PTM disrupted	166
Uncertain with motif but not PTM disrupted	782
Total with motif but not PTM disrupted	1457
Total C-terminal motifs	9165
Motifs mutated by a C-terminal variant	1258

**Table 3 biomolecules-13-00355-t003:** Variants discovered in our bioinformatics analysis.

Protein	Disease	Pathogenicity	Mutation	Motif	PTM	Frequency	GERP Score	Type
AR	Androgen resistance syndrome	Likely pathogenic	Y915S	KVKPIYFHTQ>	Phosphotyrosine	<7.95 × 10^6^	5.18	New hypothesis
APOC-III	NULL ^1^	NULL ^1^	T74A	PEVRPTSAVAA>	O-glycosylation	2.78 × 10^5^	−0.501	New hypothesis
APOE	NULL ^2^	NULL ^2^	S296R	TSAAPVPSDNH>	O-glycosylation; phosphoserine	9.48 × 10^5^	−0.0552	Unlikely
CRYM	Congenital sensorineural deafness	Pathogenic	K314T	YDSWSSGK>	Acetylation	<7.95 × 10^6^	5.45	New hypothesis
CSF1R	Hematologic neoplasm	Likely pathogenic	Y969F	LQPNNYQFC>	Phosphotyrosine	<7.95 × 10^6^	5.25	Rediscovery
CSF1R	Hematologic neoplasm	Likely pathogenic	Y969H	LQPNNYQFC>	Phosphotyrosine	<7.95 × 10^6^	5.25	Rediscovery
CSF1R	Hematologic neoplasm	Likely pathogenic	Y969C	LQPNNYQFC>	Phosphotyrosine	<7.95 × 10^6^	5.25	Rediscovery
FUS	Juvenile amyotrophic lateral sclerosis	Pathogenic	Y526C	QDRRERPY>	Phosphotyrosine	3.98 × 10^6^	0.158	Rediscovery
GluN2A	Landau-Kleffner syndrome	Likely pathogenic	S1459G	RVYKKMPSIESDV>	Phosphoserine	<7.95 × 10^6^	5.79	Rediscovery
GMPPB	Muscular dystrophy-dystroglycanopathy	Likely pathogenic	R357H	GESVPEPRIIM>	Methylation	3.72 × 10^5^	5.24	New hypothesis
HBA	Familial erythrocytosis	Pathogenic	K139E	VSTVLTSKYR>	Acetylation	<7.95 × 10^6^	4.43	Unlikely
HBA	NULL ^3^	NULL ^3^	Y140H	VSTVLTSKYR>	Phosphotyrosine	<7.95 × 10^6^	2.8	Unlikely
HBB	Familial erythrocytosis	Pathogenic	K144N	VANALAHKYH>	Acetylation	<7.95 × 10^6^	−2.8	Unlikely
HBB	Familial erythrocytosis	Pathogenic	Y145H	VANALAHKYH>	Phosphotyrosine	<7.95 × 10^6^	4.68	New hypothesis
HBB	Familial erythrocytosis	Pathogenic	Y145C	VANALAHKYH>	Phosphotyrosine	<7.95 × 10^6^	4.68	New hypothesis
HBB	Familial erythrocytosis	Pathogenic	Y145N	VANALAHKYH>	Phosphotyrosine	<7.95 × 10^6^	4.68	New hypothesis
INS	Permanent neonatal diabetes mellitus	Pathogenic	Y108C	SLYQLENYCN>	Phosphotyrosine	<7.95 × 10^6^	3.58	Unlikely
RHO	Retinitis pigmentosa	Likely pathogenic	S343N	SKTETSQVAPA>	Phosphoserine; dephosphorylation	<7.95 × 10^6^	5.42	Rediscovery
SDHA	Mitochondrial complex II deficiency; paragangliomas 5	Likely pathogenic; uncertain significance	R662C	ATVPPAIRSY>	Methylation	<7.95 × 10^6^	4.12	New hypothesis
SMAD3	Familial thoracic aneurysm and aortic dissection	Likely pathogenic	S423N	SPSIRCSSVS>	Phosphoserine	<7.95 × 10^6^	4.97	New hypothesis

^1^ Labeled in ClinVar as pathogenic for “Apolipoprotein C-III, non-glycosylated”. ^2^ Labeled in ClinVar as pathogenic for “APOE4(+)”. ^3^ Labeled in ClinVar as pathogenic for erythrocytosis but contradicted by the report identifying the mutation [30].

## Data Availability

The data analyzed in this study can be obtained from Minimotif Miner (http://mnm.engr.uconn.edu/MNM/), the C-terminome database (http://cterminome.bio-toolkit.com/), NCBI ClinVar (https://ftp.ncbi.nlm.nih.gov/pub/clinvar/xml/), UniProtKB/SwissProt (https://www.uniprot.org/help/downloads), dbNSFP4.2 (https://sites.google.com/site/jpopgen/dbNSFP?pli=1), gnomAD v2 Exomes (https://gnomad.broadinstitute.org/downloads), NCBI RefSeq (https://ftp.ncbi.nlm.nih.gov/refseq/H_sapiens/), and NCBI HomoloGene (https://ftp.ncbi.nih.gov/pub/HomoloGene/) (all accessed on 29 November 2022).

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
