# Peer review of "Systematic Assessment of Protein C-Termini Mutated in Human Disorders"

_biomolecules, 2023, doi:10.3390/biom13020355_

Round 1

Reviewer 1 Report

FitzHugh and Schiller presented an interesting study on a systematic assessment of ClinVar pathogenic variants and their potential link to C-termini post-translational modification driven disease-causing mechanisms. The manuscript is overall well-written, and the methods and results are thoroughly and clearly described. I appreciate the authors’ effort in making graphic demonstration of each novel hypothesis originated from their study. However, I have some major/minor comments listed below which I feel should be addressed before it’s ready for publication.

Major

1.     The title is a bit misleading to me, it shouldn’t be stated as “exome-wide analysis” when there is no data analysis done, would something like “systematic assessment of protein C-termini mutations in human disorders” be more appropriate?

2.     Methods: have the authors considered other disease-causing mutation databases such as HGMD or OMIM to make their variant list more complete?

3.     Table 3, it would be great if the authors can also include chromosomal positions and alleles information for these variants for future cross-reference. I am also curious how these variants have been predicted using computational algorithm-based scoring methods such as CADD or MPC? Or is it feasible at all?

4.     In general, there are very few pathogenic mutations at the C-termina identified, I am wondering how much this deviates from our expectation given it’s a small domain, what would be the expected number of mutations if we take sequence length into consideration? Comparing to other protein domains? Any biological interpretations?

5.     Line 225, if previous reports (Ref. 61, 62) identified AR Y915 site affects ligand binding and its regulation by tyrosine-phosphorylation, why is the discovery here still considered as “new hypothesis”?

6.     In general, the authors started by looking into ClinVar reported variants, as pointed out in the Discussion, these might not be comprehensive. I am curious if the look-ups can be expanded in a way beyond known pathogenic mutations, such as annotating all gnomad identified C-termini variants which might be predicted as deleterious based on a PTM-related mechanism but have not yet been reported as pathogenic previously?

Minor

1.     Line 92, ref. 151, reference numbering doesn’t seem to appear in order

2.     Table 3, it would be great if the authors can also include chromosomal positions and alleles information for these variants for future cross-reference

3.     Line 151, the font looks different

4.     Line 151, “CSF1R is causative of these tumors”, is there references to support this besides ClinVar listing? Please include if so.

5.     Line 502, format of the 1st paragraph of the Discussion is a bit strange, the authors can number the bullet points and still put them in the same paragraph.

Author Response

See the enclosed file.

Reviewer 2 Report

The manuscript by Fitz Hugh and Schiller entitled "Exome-wide analysis of protein C-termini mutated in human disorders" describes a bioinformatics approach to identify minimotifs in combination with mutated PTM sites. The authors used various databases and bioinformatics tools and identified 20 potentially disease-causing variants. For nine candidate sites, the authors hypothesized that they may have a novel function and highlighted possible mechanisms in Figure 2-8.

Unfortunately, the MnM and C-terminome database are currently unavailable, making evaluation difficult. Furthermore, other users cannot benefit from these databases and check the results. All findings of this study are highly speculative and there are no experiments to validate the results. Wouldn't it be useful if the authors collaborated with a molecular biology group to validate at least 1-2 of their results? At this point, the results are unfortunately very vague and due to the high number of PTMs with unknown function, the postulated graphs are also highly speculative. For example, the authors speculated that the K314T mutation of µ-crystallin might be associated with non-syndromic sensorineural deafness. Previous studies have found that K314 is acetylated, but why do the authors suggest that acetylation is critical for the disease? Could it also be that another PTM, including methylation or ubiquitination, is responsible for the disease? If the K314T mutation is functional, a simple co-immunoprecipitation or binding assay could confirm the results. Similarly, a kinase assay could confirm whether Src is the kinase responsible for the androgen receptor Y915. receptor. Overall, the data are interesting but without further validation, the manuscript is in its current form descriptive and highly speculative.  

Minor:

Abstract: What means “not very frequent”?

Page 21 line 508: what means “less common”?

Have the authors also used other databases for human diseases and variants, such as ClinGen or TopMed?      

Author Response

See enclosed file.

Round 2

Reviewer 1 Report

The authors have addressed all of my questions and concerns in the revised manuscript, I support the publication in its current status.

Author Response

Many thanks for your kind review.